# Ligand-induced conformational rearrangements regulate the switch between membrane-proximal and distal functions of Rho kinase 2

István Hajdú [1✉], András Szilágyi [1], Barbara M. Végh [1,2], András Wacha [3], Dániel Györffy[1,4], Éva Gráczer [1], Márk Somogyi[1], Péter Gál[1] & Péter Závodszky [1✉]

Rho-associated protein kinase 2 (ROCK2) is a membrane-anchored, long, flexible, multi-domain, multifunctional protein. Its functions can be divided into two categories: membrane-proximal and membrane-distal. A recent study concluded that membrane-distal functions require the fully extended conformation, and this conclusion was supported by electron microscopy. The present solution small-angle X-ray scattering (SAXS) study revealed that ROCK2 population is a dynamic mixture of folded and partially extended conformers. Binding of RhoA to the coiled-coil domain shifts the equilibrium towards the partially extended state. Enzyme activity measurements suggest that the binding of natural protein substrates to the kinase domain breaks up the interaction between the N-terminal kinase and C-terminal regulatory domains, but smaller substrate analogues do not. The present study reveals the dynamic behaviour of this long, dimeric molecule in solution, and our structural model provides a mechanistic explanation for a set of membrane-proximal functions while allowing for the existence of an extended conformation in the case of membrane-distal functions.

[1] Institute of Enzymology, Research Centre for Natural Sciences, Budapest, Hungary. [2] ELTE NAP Neuroimmunology Research Group, Department of Biochemistry, Institute of Biology, ELTE Eötvös Loránd University, Budapest, Hungary. [3] Institute of Materials and Environmental Chemistry, Research Centre for Natural Sciences, Budapest, Hungary. [4] Faculty of Information Technology and Bionics, Pázmány Péter Catholic University, Budapest, Hungary. ✉email: hajdu.istvan@ttk.hu; zavodszky.peter@ttk.hu

Rho-associated coiled-coil containing kinases (ROCK1 and ROCK2) are known to have multiple repertoires of activities in several cellular functions[1]. Their downstream targets are involved in cell shape and motility, cell survival and apoptosis, vesicle dynamics, cell growth and regeneration, and cytoskeleton regulation. Their most prominent targets are membrane-distal, localized at the actomyosin filaments, and include LIM kinases[2], the myosin phosphatase target subunit (MYPT) of myosin light chain phosphatase (MLCP)[3], myosin light chain II (MLC-2)[4], collapsin response mediator protein 2 (CRMP2)[5], and ERM proteins[6] (ezrin, radixin, and moesin). ROCK (particularly ROCK2) kinases are associated with numerous neurodegenerative diseases[1] where not all of the mechanisms are identified, however, the few known targets are membrane-proximal proteins (Amyloid Precursor Protein, beta secretase-1[7]). ROCK kinases are localized mainly in the cytoplasm[8] with their C-terminal domains anchored to the cell membrane[9,10]. As the locations of the downstream targets differ by as much as 100 nm, the conformation of the active ROCK proteins must sample a large conformational space.

The human ROCK1 and ROCK2 are highly homologous kinases consisting of 1354 and 1388 amino acids, respectively. Both ROCK proteins were identified in the 1990s[4,11,12]. The two ROCK proteins share 64% identity in their amino acid sequence with the highest (92%) identity in their kinase domains[13]. The functions of these kinases were initially considered highly overlapping, but knockout studies showed that disruption of either gene is sufficient to generate stillborn embryos[14,15], indicating that at least in early development, the two enzymes cannot substitute for each other, although it was shown that they can compensate for each other in adult organisms[16]. Both enzymes are serine/threonine kinases with a similar overall structure. The catalytic (kinase) domains are located at their N-terminus. The center of the molecule forms a long coiled-coil domain containing the Rho-binding domain. The C-terminal region contains a split pleckstrin homology domain, which is bisected by an internal cysteine-rich zinc-finger domain. The enzymatic activity of ROCK kinases is strongly enhanced by homodimerization[17] where the N- and C-terminal extensions of the kinase domain function as a dimerization interface in addition to the coiled-coil helical regions. The dimerization of the protein is suggested to play a role in its biological function towards dimeric substrates[13].

In their native form, ROCK kinases are regarded as enzymatically inactive owing to auto-inhibition of the kinase domain by the C-terminal region[18,19]. The auto-inhibition model is strongly supported by the finding that deletion of the C-terminal region leads to a constitutive activation of the kinase. The auto-inhibitory region is permanently cleaved during apoptosis by caspase-3 for ROCK1[20,21] and granzyme-B for ROCK2[22] to create active ROCK kinases leading to actin-myosin contraction, membrane blebbing, and formation of apoptotic bodies. The auto-inhibition model is also supported by SPR studies showing binding between the isolated kinase and PH domains of ROCK1[23].

An intriguing question remaining open is related to the Rho-dependent activation of ROCK kinases. The traditional view of the activation states that the "auto-inhibited" form of ROCK is inactive, and binding of various Rho (RhoA and RhoC) proteins causes a disruption of the inhibited conformation, thus ROCK kinases become active as a result of a conformational change[24]. This view is challenged by the fact that several experiments[4,8,17,19] have shown that the "auto-inhibited" form is active as well, and the Rho-driven activation leads only to a 1.5–3-fold activity increase as found on various substrates.

The structures of the individual domains of Rho kinases are known[13,25–28], but no high-resolution structure of the full-length protein is available. Truebestein et al.[29] determined the low-resolution structure of ROCK2 using electron microscopy, showing an extended coiled-coil region serving as a "molecular ruler", keeping the N- and C-terminal domains 110 nm apart from each other. This finding further challenged the Rho-driven activation theory of ROCK2, and while it provided an explanation for the membrane-distal functions, it left the membrane-proximal functions unexplained.

Our aim was to determine the solution conformation of ROCK2 alone and complexed with RhoA in order to understand the activation mechanism and find structural explanation for the dual sets of functions—membrane-proximal and membrane-distal. Our approach was small-angle X-ray scattering combined with computer modeling and functional studies, involving the effect of RhoA binding on ROCK2 activity, and the inhibitory effect of the C-terminal domains.

## Results

**Interaction of ROCK2 with RhoA as supported by size exclusion chromatography.** Although ROCK2 kinase was initially identified as a Rho-associated protein, the direct binding between recombinant RhoA and ROCK2 was recently questioned[29], despite the fact that even a crystal structure of a complex between RhoA and a ROCK1 fragment had been published[25]. To verify the existence of the complex between GTP-bound activated RhoA and full-length ROCK2, we performed size exclusion chromatography on a mixture of ROCK2 and RhoA labeled with the fluorescent GTP analog BODIPY. BODIPY-GTP was also mixed with ROCK2 or RhoA alone. Although we could not detect the ternary complex between ROCK2/RhoA/BODIPY-GTP directly, indirect evidence was obtained: BODIPY-GTP bound much more strongly to RhoA in the presence of ROCK2 than in its absence, suggesting the presence of a ternary complex similar to that measured for ROCK1[30], although this ternary complex is weak, and the dynamic equilibrium between ROCK2 and RhoA/BODIPY-GTP is probably shifted towards the dissociated form. (Figs. 1 and S1).

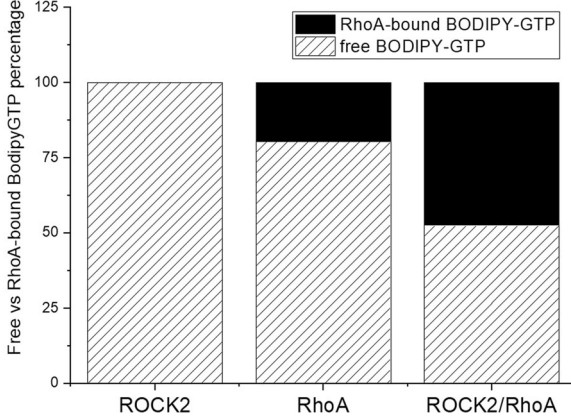

**Fig. 1 Formation of the ROCK2/RhoA/BODIPY-GTP complex measured by size exclusion chromatography.** Mixtures of 5 μM ROCK2 and 5 μM BODIPY-GTP (left); 5 μM RhoA and 5 μM BODIPY-GTP (middle) 5 μM ROCK2, 5 μM RhoA, and 5 μM BODIPY-GTP (right) was run and followed at 280 nm (protein) and 488 nm (BODIPY fluorophore) absorbance. When BODIPY-GTP is used in a binary mix with ROCK2, BODIPY-GTP is quantitatively in the free form (dashed surface). In the binary mix with RhoA 20% of BODIPY-GTP is in complex (solid black surface) with RhoA. In the ternary mix, 48% of BODIPY-GTP is in complex with RhoA.

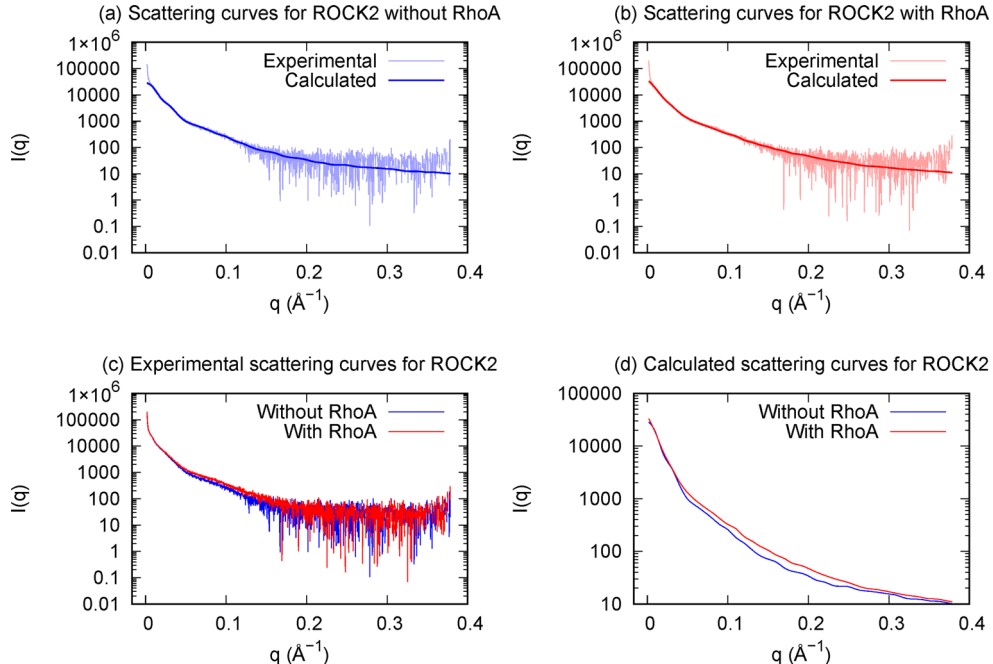

**Fig. 2 Experimental and calculated SAXS-scattering curves for ROCK2 with and without bound RhoA.** Calculated curves were computed from the best model obtained during geometrical simulation. **a** Experimental and calculated curves for ROCK2 without RhoA. **b** Experimental and calculated curves for ROCK2 with bound RhoA. **c** Comparison of experimental curves for ROCK2 with and without bound RhoA. **d** Comparison of calculated curves for ROCK2 with and without bound RhoA.

**RhoA binding to ROCK2 as detected by SAXS measurements**. Before the small-angle X-ray scattering (SAXS) measurements, we determined the highest concentration of ROCK2 that can be maintained without aggregation (Fig. S2) in order to obtain the best possible scattering signals. The highest achievable concentration was 5.4 μM for MBP-ROCK2 and 3.8 μM for ROCK2. For our further studies, we used these concentrations as the starting point for making serial dilutions. For the RhoA-bound state, 1.5× molar excess of GppNHp-bound RhoA was added to the ROCK2 forms.

Solutions of ROCK2 protein in the absence and presence of RhoA were investigated using synchrotron SAXS to gain insight into the overall structure of the protein. Guinier analysis was performed on the scattering profiles, yielding the radii of gyration ($R_g$) and the molecular mass of the proteins, with the latter based on the scattering intensity extrapolated to zero angle (Fig. S3). As scattering curves were noisy for the MBP-free construct, we used the scattering results obtained with the fusion variant for further analysis. The $R_g$ values were $14.63 \pm 0.18$ nm for the MBP-ROCK2 alone, and $15.09 \pm 0.16$ nm for MBP-ROCK2 in the presence of RhoA, indicating some loosening of the structure. The experimental molecular masses (458 and 490 kDa for MBP-ROCK2 and MBP-ROCK2/RhoA, respectively) agreed with the calculated masses of the dimeric MBP-ROCK2 (408 and 450 kDa) within the margin of experimental error. The Kratky plots indicated mostly folded, somewhat flexible structures for both protein forms. This agrees well with the hypothesized coiled-coil structure, being well-ordered (helical) on the atomic scale while more flexible on larger size ranges. To interpret the differences between the scattering curves, we performed molecular modeling of the ROCK2 structure.

**ROCK2 structural models from SAXS**. We have obtained a good fit between the experimental scattering curves and those calculated from the best models obtained by geometric simulations (Fig. 2). In the best model we have obtained for the MBP-

ROCK2 fusion protein without bound RhoA, the chain is folded in half and one N-terminal domain is in contact with a C-terminal domain (Fig. 3a). In the best model obtained for the RhoA-bound MBP-ROCK2, the molecule is less compact ($R_g$: 15.8/11.8 nm with/without RhoA), in agreement with the experimentally detected increase of $R_g$ complexed with RhoA, and the termini are not in proximity (Fig. 3b). The discrepancies between the Rg values obtained from the scattering data and those calculated from the structural models are due to the fact that the experimental Rg is calculated from a small stretch of the Guinier plot while the structural models are scored based on the whole scattering curve; also, the experimental value is for a solution ensemble of structures while the value obtained from the structural model is for a single structure. Analysis of the structural ensembles generated for both molecules supports the conclusion that RhoA binding causes a population shift towards structures with a higher radius of gyration and a longer distance between the termini (Fig. 4).

**Enzyme activity on natural and small synthetic substrates**. To test the auto-inhibitory function of the C-terminal domains of ROCK2, we expressed the kinase domain with the extension required for dimerization (1–420 aa) and compared its activity with that of the full-length ROCK2 on three large natural substrates (LIM kinase 1 and 2, MYPT1) and two small synthetic peptides (provided with the Z'Lyte Thr/Ser 07 and 13 kits). The results are presented in Fig. 5 and Table 1.

On LIMK1, the activity of the full-length ROCK2 is 56% of that of the isolated kinase domain, while on the myosin binding subunit (MYPT1) of myosin phosphatase and LIMK2, the measured activity of the full-length kinase was higher (by 19% and 11%, respectively) than that of the kinase domain. This difference is close to the standard deviation of the experiment, so we can make the qualitative statement that no apparent inhibition occurs in the presence of the C-terminal domains. On the small

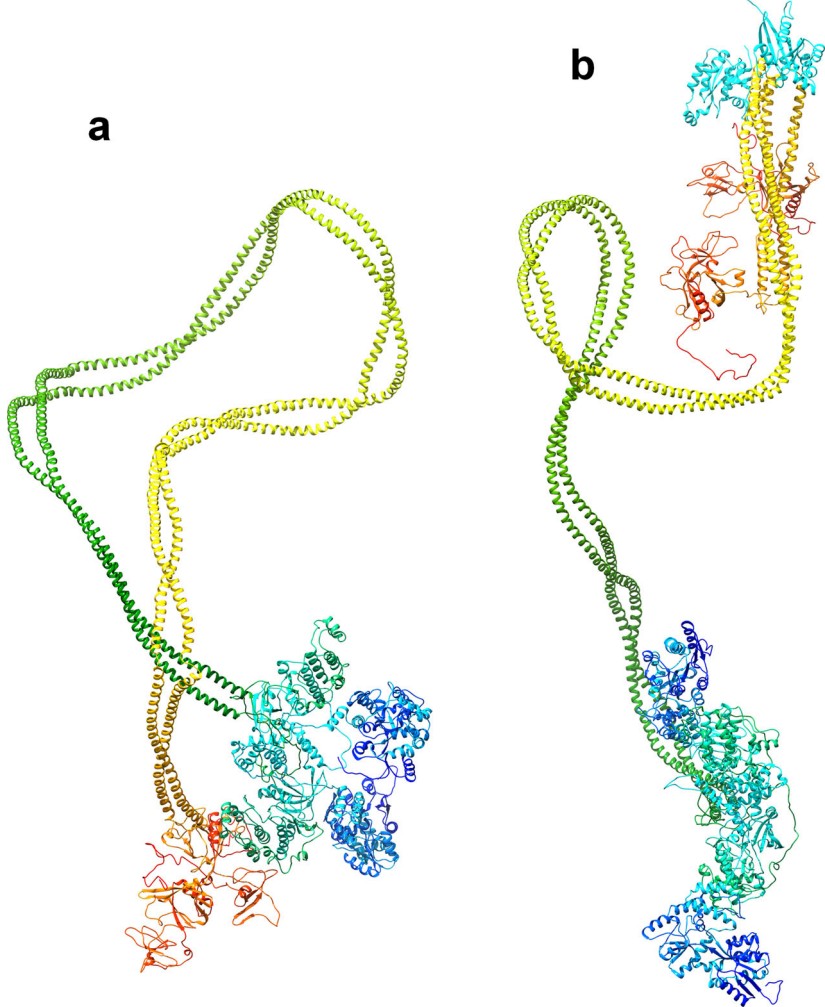

**Fig. 3 Solution structure of full-length ROCK2.** Structure of the best model obtained for full-length ROCK2 without (**a**) and with (**b**) bound RhoA protein. RhoA subunits are shown in cyan.

peptide substrates, the activity of the full-length ROCK2 was measured to be 14% percent of that of the isolated kinase domain.

To test the RhoA induced activation, we repeated the activity measurements with RhoA(GppNHp)-complexed with full-length ROCK2, and observed a slight 36% increase in enzyme activity upon addition of RhoA in the case of Z'Lyte substrates compared to the full-length kinase. In the case of natural substrates, the presence of RhoA resulted in a very small enzyme activity change (3% decrease for LIMK1, 12% decrease for LIMK2 and 9% increase for MYPT1) compared with the RhoA-free variant.

## Discussion

The activation mechanism of ROCK kinases has long been regarded as a binding-induced conformational change where the C-terminal region of ROCK acts as an auto-inhibitory module, by binding to the N-terminal kinase domain and inhibiting its activity[24], whereas the binding of GTP-bound Rho proteins on the coiled-coil region of ROCK induces a conformational change disrupting the binding between the kinase domain and the C-terminal region. However, no structural data are available to support this proposed mechanism. The main reason for this gap in our knowledge lies in the nature of ROCK kinases. These proteins are huge (2 × 1354 and 2 × 1388 aa for ROCK1 and ROCK2, respectively), and hard to express in heterologous recombinant systems. Using our recently developed unified expression system (pONE)[31], we successfully expressed the full-

size, dimeric ROCK2 protein with an MBP tag in insect cells, in sufficient amounts required for structural studies.

The solution structures of the RhoA-free and Rho-bound forms were investigated using SAXS. To interpret the experimental scattering curves shown in Fig. 2, molecular simulation studies were conducted. We concluded that ROCK2 in the RhoA-free state can be represented by a distribution of structures dominated by such conformations where the N-terminal kinase and the C-terminal regulatory domains are in proximity (as illustrated in Fig. 3a). This result is in concert with the auto-inhibitory function. The set of structures of ROCK2 complexed with RhoA is shifted towards conformations, where the N- and C-terminal domains are not in contact (Fig. 3b). The simulation also suggests enhanced conformational flexibility of the ROCK2-RhoA complex compared to the RhoA-free form (Fig. 4) in accord with an earlier assumption[24]. Our results offer a mechanism for the structural interpretation of ROCK2 regulation, showing that RhoA induces a shift in the conformational ensemble of ROCK2 towards an extended, open and more flexible set. This dynamic model can explain both membrane-proximal and membrane-distal functions, and provides structural interpretation for the regulatory effect of the substrates.

A recent model based on electron microscopy suggested that ROCK2 is a semi-rigid rod with the N- and C-terminal domains at a large distance from each other[29,32]. The experimental conditions used for the electron microscopy may have shifted the

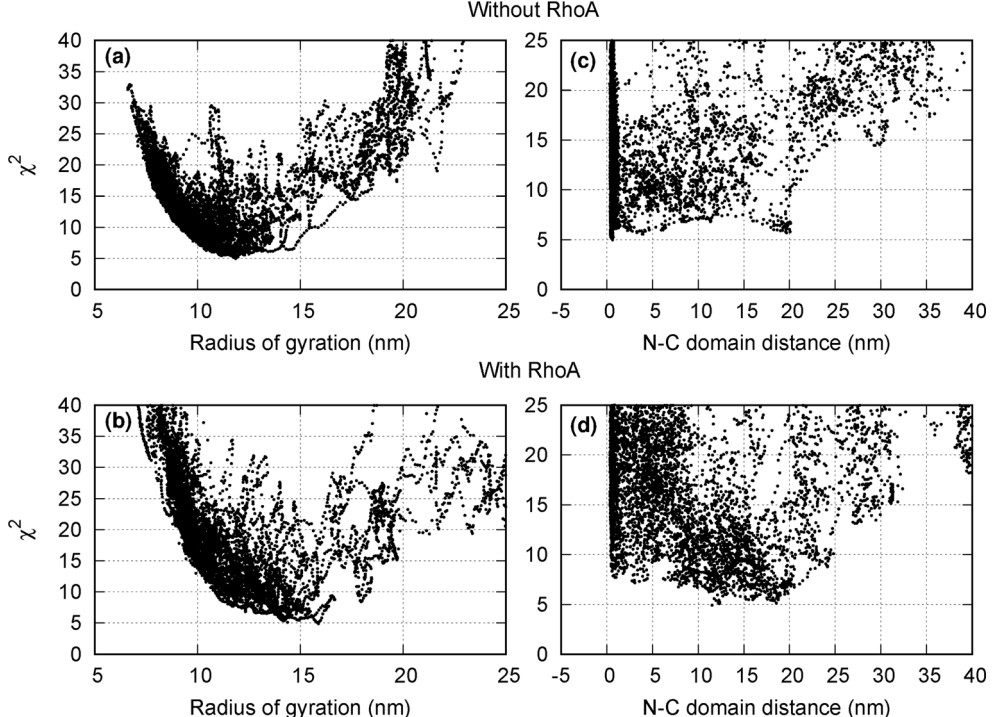

**Fig. 4 Deviation between the experimental SAXS-scattering curve and the curves calculated for structures generated by geometric simulation, expressed by the reduced chi-squared ($\chi^2$) value.** The graphs in the left column **a** and **b** show the deviation as a function of the radius of gyration; those in the right column **c** and **d** show it as a function of the distance between the N- and C-terminal domains. The top row **a** and **c** shows graphs for ROCK2 without bound RhoA; the bottom graphs **b** and **d** are for ROCK2 with bound RhoA. The structures with the lowest $\chi^2$ values tend to have a higher radius of gyration and longer distance between the termini in the RhoA-bound than in the RhoA-free species.

population towards the extended conformations, although it should be noted that the electron microscopic images also show a number of more compact states (see Fig. 1d in ref. [29]). Our simulations suggest that fully extended conformations are entropically unfavorable, and therefore have low probability in solution. Our SAXS measurements show that Rho-binding to ROCK2 induces a population shift towards open conformation, capable of forming heterologous interactions with various partners rather than intramolecular contacts. Numerous interaction partners (myosin light chain phosphatase, ERM proteins, LIM kinases) of ROCK kinases are parts of the cytoskeleton and are located ~120 nm beneath the cell membrane. According to our integrated model of activation (Fig. 6), the ROCK molecule is anchored to the membrane at its C-terminus, and the N-terminal kinase domain stays in its proximity; the flexible coiled-coil domain allows such freedom. Upon the interactions with Rho proteins, the average distance of the kinase domain from the membrane increases, the kinase domain no longer remains in close proximity of the C-terminal domains; consequently the membrane binding region does not influence its activity directly. Owing to the increased flexibility, the kinase domain becomes capable of forming heterologous interactions with various partners. Not only RhoA, but also the substrates of ROCK2 are probably capable of promoting the adoption of the extended conformation by forming protein–protein interactions with the loose closed-conformation forms of ROCK2. When in complex with a cytoskeletal partner, the coiled-coil can be stretched to offer ROCK2 the role in mediation of actomyosin contraction.

Our unified model based on conformational flexibility can also explain the membrane-adjacent activities of ROCK2 as kinase activity does not require an extended conformation. In the RhoA-free state, the N-terminus is localized near the C-terminus, thus near the plasma membrane. Our experiments show that the self-associated weak complex can be dissociated by natural substrates, giving access of the substrate to the kinase domain. Our comparative activity measurements obtained with the full-length ROCK2 and its isolated kinase domain showed that the nature of the substrate makes a difference (Fig. 5). While on small peptide substrates, the presence of the C-terminal domains of ROCK2 significantly reduces the kinase activity (to 13–15% Fig. 5a, b), on natural protein substrates (LIM kinase 1, LIM kinase 2, and MYPT1 Fig. 5c–e), the activities of the full-length ROCK2 and its isolated kinase domain do not differ significantly. Our interpretation of this phenomenon is that large, multidomain natural substrates access the kinase domain of ROCK2 by inducing the necessary conformational rearrangement, while small synthetic substrates are not capable of doing this. We assume the existence of allosteric binding site(s) on ROCK2, which facilitate the specific selection of natural substrates.

Another interesting observation was that RhoA does not influence ROCK2 activity, irrelevant of the size and nature of substrates. The observed increase in the $R_g$ values upon RhoA binding can be explained as a shift in the dynamic equilibrium between closed and partly open conformations. Based on our SAXS and enzyme kinetic results, we can clearly state that the function of RhoA is not to activate the ROCK2 kinase, just to enable the kinase domain to access its membrane-distal substrates.

## Methods

### Protein expression and purification

*ROCK2, ROCK2-KD constructs, expression, and purification*. The synthetic gene of human ROCK2 (1–1388) was cloned into pONE30A[31] vector between AvrII and NotI cleavage sites. The ROCK2 kinase domain (1–420) construct (ROCK2-KD) truncation was prepared using primers 5′-GATCGCTAGCCCTAGGGGATCC-3′ and 5′-GATCGCGGCCGCTTCGCGGCAGGAGG-3′, cloned between NheI and NotI restriction sites. Both protein constructs (containing MBP tag at the

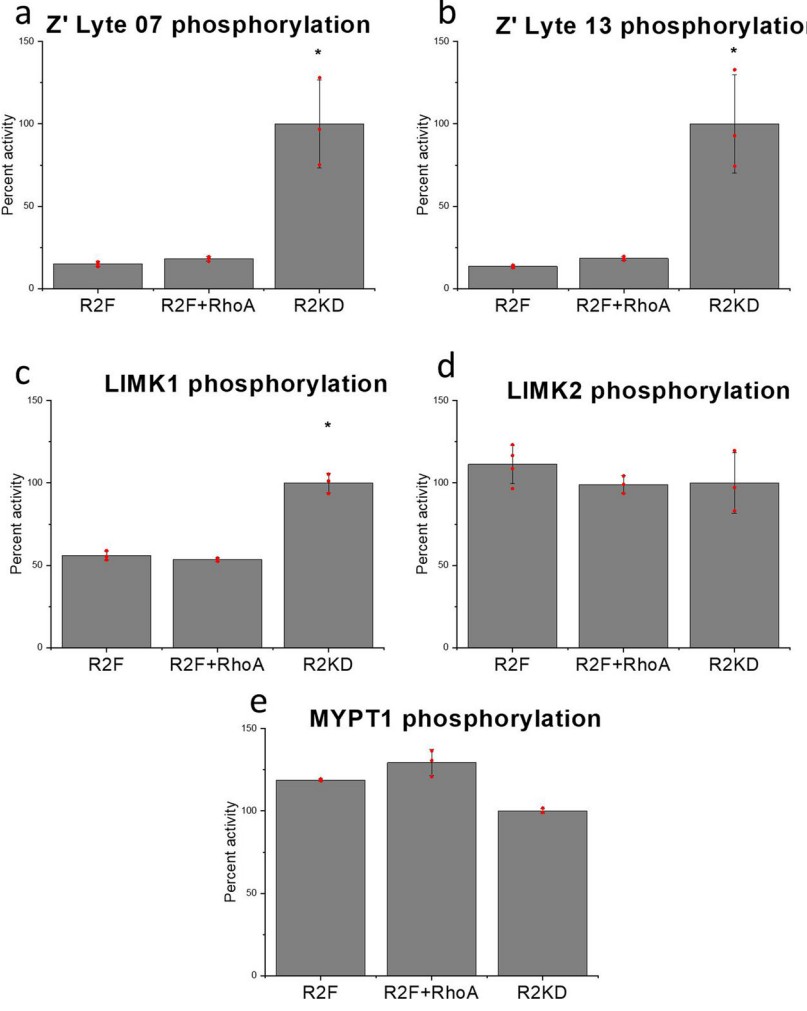

**Fig. 5 ROCK2 activities on different substrates.** The activities of full-size ROCK2 (R2F) and its kinase domain (R2KD) were analyzed using natural (LIMK1, LIMK2, MYPT1) and synthetic substrates. **a** The phosphorylation of Z'Lyte Ser/Thr 07 **a** and 13 **b** peptide were measured in 384-well plates. The calculation of relative activities was based on enzymatic activity at a fixed peptide and saturating ATP concentrations. The enzyme activity of ROCK2 was inhibited to 14% of that of the ROCK2 kinase domain, and the presence of RhoA increased it to 19% percent. **c** The phosphorylation of LIMK1 was followed by the incorporation of 32P-ATP. Samples were taken at different time-points, run on a SDS-PAGE gel and analyzed on a phosphorimaging screen. Relative phosphorylation of the substrates using different enzymes was calculated from the time course of phosphorylation. The enzyme activity of ROCK2 on LIMK1 was half of that of the ROCK2 kinase domain, and RhoA binding did not increase it. The enzyme activity of ROCK2 was practically the same (111% for LIMK2, **d** and 119% for MYPT1, **e**) as that of the kinase domain. The addition of RhoA had no significant effect on the activity of ROCK2. The error bars represent the standard deviation of at least three independent experiments in all cases. The $p$ values were calculated from a two-sided $t$ test. *$p < 0.05$ vs. full-size ROCK2.

N-terminus) were expressed in Sf9 insect cells after cotransfection using flashBAC GOLD baculoviral expression system (Oxford Expression Technologies). After virus amplification, insect cells at $2 \times 10^6$ cell/ml density were transfected with the recombinant baculovirus to express the protein of interest in 500 mL Insect-XPress medium (Lonza), for 3 days at 27 °C, in a shaker flask at 220 rpm. The proteins were purified using amylose affinity chromatography in 25 mM HEPES pH 7.4, 500 mM NaCl and 1 mM dthiothreitol (DTT). The bound proteins were eluted by buffer containing 10 mM maltose. The collected protein fractions were dialyzed and concentrated against 25 mM HEPES buffer (pH 7.4).

For enzyme activity and size exclusion chromatography measurements the MBP tag was removed by tobacco etch virus protease cleavage. The cleavage was performed in 15 times molar excess of TEV at room temperature for 2 h. The His-tagged ROCK2 constructs were separated from MBP using Ni-affinity chromatography. The eluted protein was applied to a Superose 6 (ROCK2-full) or Superdex 75 (ROCK2-KD) gel filtration column equilibrated in 25 mM HEPES pH 7.4, 150 mM NaCl, 1 mM DTT. Peak fractions were pooled and concentrated. The purity of proteins was tested by sodium dodecyl sulphate–polyacrylamide gel electrophoresis (SDS-PAGE). stained with Coomassie Brilliant Blue. ROCK2-full and ROCK2-KD proteins were concentrated and stored at −80 °C.

*RhoA construct, expression, and purification.* The synthetic gene of the human RhoA (1–181) was cloned into a pONE10K vector[31] between NcoI-NotI cleavage

sites. The construct was transformed into Rosetta2 *E. coli* cells and the protein (containing additional AS- sequence at the N-terminus and the -AAHHHHHH sequence at the C-terminus) was expressed at 37 °C in the presence of 50 μg/mL kanamycin and 30 μg/mL chloramphenicol. The RhoA gene expression was induced with 1 mM isopropyl β-d-1-thiogalactopyranoside. The cells were suspended in 25 mM HEPES pH 8.0, 300 mM NaCl, 20 mM imidazole, 1 mM DTT and Complete Ultra protease inhibitor were also added to the solution. The His-tagged RhoA(1–181) protein was purified using Ni+ affinity chromatography. The bound RhoA protein was eluted by buffer containing 250 mM imidazole. The collected protein fractions were dialyzed against 25 mM HEPES, 150 mM NaCl, 1 mM DTT buffer (pH 7.4). The dialyzed enzyme solution was frozen in liquid nitrogen and stored at −80 °C.

*LIM kinase constructs, expression, and purification.* LIMK1 kinase domain (330–647) and full sized LIMK2 was expressed as an MBP-fusion protein in baculovirus-insect cell expression system. The LIM kinase coding genes were cloned into pONE30A vector for cotransfection. After virus amplification, insect cells at $2 \times 10^6$ cell/ml density were transfected with the recombinant baculovirus to express the protein of interest for 3 days at 27 °C, at a constant shake of 220 rpm. The cells were centrifuged and resuspended in 25 mM HEPES, 500 mM NaCl, pH 7.4 buffer containing 1 mM phenylmethylsulfonyl fluoride (PMSF) and 1 mM DTT, then sonicated. Cell debris were centrifuged and the MBP-LIM kinases was

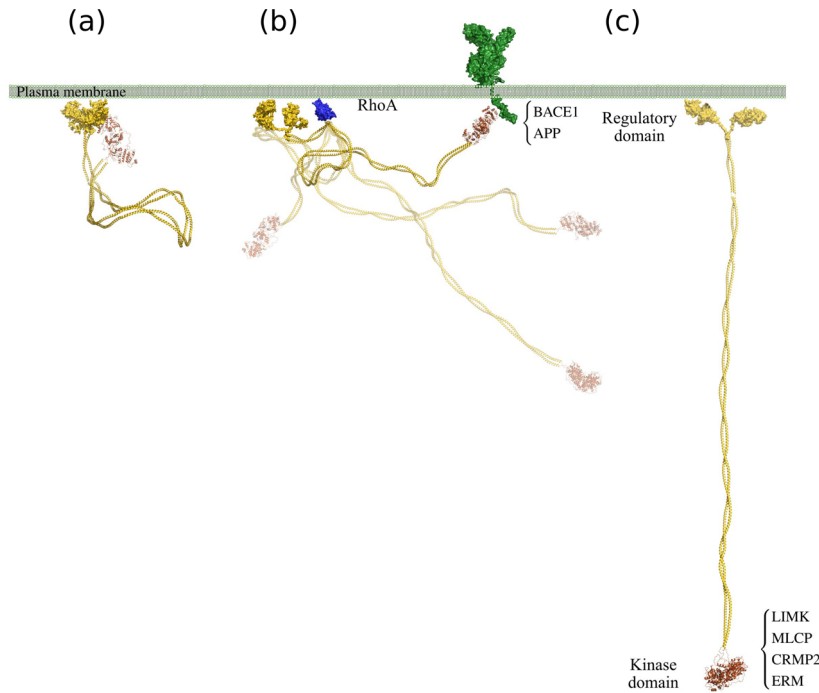

**Fig. 6 Model for ROCK2 activation.** ROCK2 is anchored to the membrane by its C-terminal domain, and the N-terminal kinase domain is folded back forming a compact conformation with limited flexibility (**a**). Since the binding between the terminal domains is weak, a substrate capable of forming a strong interaction with the kinase domain can displace the C-terminal domains. Binding of RhoA induces transient loosening of the compact structure, and releases the N-terminal part of the molecule **b**, allowing interactions with substrates at distant locations, e.g., the cytoskeleton. Such interactions stabilize the extended conformation of ROCK2 (**c**).

**Table 1 Phosphorylation rate ($V_{meas}$) of different substrates by ROCK2 enzyme variants measured at low substrate concentrations, in $\text{min}^{-1}$ units.**

| Substrate | ROCK2 full length | ROCK2 full length + RhoA | ROCK2 kinase domain |
|---|---|---|---|
| Z' Lyte 07 | 1.44 ± 0.14 (15%) | 1.74 ± 0.14 (18%) | 9.61 ± 2.56 (100%) |
| Z' Lyte 13 | 1.37 ± 0.08 (13%) | 1.87 ± 0.12 (18%) | 10.22 ± 3.37 (100%) |
| LIMK1 | 0.081 ± 0.004 (56%) | 0.077 ± 0.001 (54%) | 0.144 ± 0.008 (100%) |
| LIMK2 | 0.430 ± 0.045 (111%) | 0.383 ± 0.021 (99%) | 0.386 ± 0.071 (100%) |
| MYPT1 | 0.181 ± 0.001 (119%) | 0.197 ± 0.012 (129%) | 0.152 ± 0.002 (100%) |

Percentages relative to the $V_{meas}$ for the isolated kinase domain are provided in parentheses.

purified from the supernatant using amylose affinity chromatography. The bound proteins were eluted by buffer containing 10 mM maltose. During affinity chromatography, peak fractions were collected and analyzed with SDS-PAGE and western blot. The collected protein fractions were dialyzed and concentrated against 25 mM HEPES buffer (pH 7.4).

Myosin phosphatase (MYPT1) expression and purification was conducted by a previously published method[33].

**Binding studies.** The ROCK2/RhoA complex formation was studied using size exclusion chromatography on a Superose 12 column on Akta Avant system in 20 mM HEPES pH 7.4, 150 mM NaCl, 3 mM MgCl$_2$, running buffer. For different runs, 5 μM RhoA, 5 μM ROCK2 and 5 μM BODIPY-GTP were used in combinations. UV absorbance was monitored at 280 and 488 nm for protein and BODIPY absorbance, respectively.

**Enzyme activity measurements.** Radioligand reactions were carried out in 50 mM HEPES, pH 7.5, 100 mM NaCl, 5 mM MgCl$_2$, 0.05% octylphenoxypolyethoxyethanol, 5% glycerol, 2 mM DTT using recombinantly expressed and purified LIMK1 (2 μM), LIMK2 (2 μM) or MYPT1 (1 μM) proteins in the presence of 400 μM ATP and ~5 μCi of [γ-32P]ATP. Reactions were stopped with protein loading sample buffer complemented with 20 mM EDTA and subjected to SDS-PAGE. Gels were dried before analysis by phosphorimaging on a Typhoon Trio+ scanner (GE Healthcare).

Fluorimetric kinase assays were performed using the Z'Lyte kit (Invitrogen) according to the manufacturer's instructions. Kinase reactions contained 400 μM ATP and 1 μM Z'Lyte 07/13 peptide substrate in 50 mM HEPES (pH 7.5), 10 mM MgCl$_2$, 1 mM EGTA, 0.01% BRIJ-35 buffer. Fluorescence was read at 445 nm (coumarin) and 520 nm (fluorescein) using 400 nm excitation wavelength on an EnSpire Multimode Plate Reader.

**Small-angle X-ray scattering.** Synchrotron X-ray solution scattering data were collected at the EMBL P12 beamline (PETRA III, DESY, Hamburg, Germany) using a robotic sample changer[34]. Initially, the data were reduced and processed using an automatic pipeline of software developed at EMBL[35]. ROCK2 was prepared in 25 mM HEPES, pH 7.4 to obtain a concentration series in the 0.3–1.1 mg/ml range. SAXS data were recorded at 25 °C using a PILATUS 2 M pixel detector at a sample-detector distance of 4 m and a wavelength of 0.124 nm, corresponding to the range 0.0019 Å$^{-1}$ < s < 3.79 Å$^{-1}$, where s, the scattering variable or momentum transfer is defined as $s = 4*\pi*\sin(\theta)/\lambda$, 2θ being the scattering angle and λ the wavelength. The software PRIMUS[36] was used for data processing. The intensity calibration was performed using the scattering from BSA at a known concentration as a secondary standard. The forward scattering $I(0)$ and $R_g$ values were determined using the Guinier approximation assuming that at very small angles ($s < 1.3\ R_g$), the intensity is represented as $I(s) = I(0) \cdot \exp(-(s\ R_g)^2/3)$. The pair-distance distribution function $P(r)$, from which the maximum particle dimension ($D_{max}$) and $R_g$ were estimated, was computed using GNOM[37]. Guinier fitting was executed with the autorg function of ATSAS 3.0.2 package[38]. The fit range was $0.00584 \leq s \leq 0.01042$ Å$^{-1}$ for

ROCK2 in the absence and $0.00521 \leq s \leq 0.01042 \text{ Å}^{-1}$ in the presence of RhoA. The molecular masses were derived from extrapolation to zero scattering angle on absolute scale.

**ROCK2 structure prediction using SAXS data**. SAXS measurement data were obtained for the full-length dimeric ROCK2 molecule (including MBP tags on the N-termini) with and without two RhoA molecules bound to the rho-binding regions. The following procedure was used to find conformations compatible with SAXS data.

*Construction of an initial model*. MARCOIL[39] was used to predict continuous coiled-coil segments and their starting registers in the ROCK2 sequence. Five such segments were identified. Structures for individual segments were constructed using the CCFold program[40]. Missing side chains were added using the Profix tool of the Jackal package[41]. The resulting coiled-coil segments were then manually roughly aligned along a line, and Modeler[42] was used to connect the segments into a continuous chain. Modeler was also used to construct the structure of the C-terminal domain from the templates 2rov (split PH domain of ROCK2 from rat) and 2row (C1 domain of the same), and that of the N-terminal domains from the MBP structure 1jw4 and the kinase domain from 4wot (the biological dimer was used). Finally, the full-length dimer ($2 \times 1792$ residues) was built, again using Modeler, by joining the N-terminal domains, the coiled-coil region, and the C-terminal domains. The RhoA protein was also modeled by Modeler from the template 1s1c, adding the His tags used in our construct. The RhoA-bound ROCK2 dimer was built by fitting the RhoA models onto the RhoA binding region of ROCK2 using the 1s1c structure, which contains the RhoA-binding region of ROCK2. The resulting ROCK2 model (with or without RhoA molecules bound) were initially in a fully extended conformation. To fix potential bad geometry and close contacts, hydrogen atoms were added and an energy minimization and a short vacuum MD simulation (50 ps) were performed with GROMACS 5.1[43] using the CHARMM27 force field.

*Conformational sampling and fitting to SAXS data*. The geometric simulation program FRODAN[44] was used to obtain a broad sampling of the vast conformational space of the full-length ROCK2 dimer. This method defines rigid substructures based on the hydrogen bonds and hydrophobic contacts as constraints, and uses momentum perturbation to generate motion along the remaining degrees of freedom. Structures were saved after every 100th step of the simulation. During the simulation, the ROCK2 molecule was observed to collapse from an extended to a rather compact conformation while sampling conformations with a wide range of compactness. To evaluate the structures with regard to the SAXS data, the CRYSOL[45] program in the ATSAS package[38] was used to generate theoretical SAXS curves, and reduced chi-squared ($\chi^2$) values were used to measure deviation from the experimental curve. In order to obtain structures with a good fit on the SAXS data, multiple threads of the simulation were started repeatedly from structures with low $\chi^2$ values. Structures with low $\chi^2$ values were obtained after about 700,000 simulation steps. Further optimization was carried out by starting 10 new simulation threads from the best structure, and recalculating the $\chi^2$ value after 100 steps; then the procedure was repeated with the best out of the 10 resulting structures until the $\chi^2$ value did not decrease further.

**Statistics and reproducibility**. The enzyme activity results represent three biological replicates. For comparisons two-sided Students' $t$ test was used. A value of $p < 0.05$ was considered statistically significant. All SAXS results are averages of 20 technical replicates. No data points were excluded from the analyzes.

**Reporting summary**. Further information on research design is available in the Nature Research Reporting Summary linked to this article.

## Data availability
The authors declare that the data supporting the findings of this study are available within the paper and its Supplementary Information files or from the corresponding author on reasonable request. The source data for the figures presented in this study have been included in Supplementary Data 1.

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

## Acknowledgements

The authors greatly thank Mária Vas (Institute of Enzymology, Research Center for Natural Sciences) for her critical review of the manuscript, Attila Bóta (Institute of Materials and Environmental Chemistry, Research Center for Natural Sciences) for his expert help in preliminary SAXS experiments. We also thank Nelly R. Hajizadeh and Dmitri I. Svergun at the EMBL SAXS facility for their professional help. The present work was supported by grants K108642, K128262, K134711, PD124451, and 2017-1.2.1-NKP-2017-00002 provided by National Research, Development and Innovation Office (NKFIH).

## Author contributions

Conceptualization: I.H., A.S., and P.Z. Formal analysis: I.H., A.S., and A.W., investigation: I.H., B.M.V., E.G., and M.S. Data curation: I.H., A.S., A.W., and D.G. Writing: I.H., A.S., and P.Z. Data visualization: I.H., A.S., and D.G. Supervision: A.S., P.G., and P.Z. Project administration: I.H., funding acquisition: P.Z.

## Competing interests

The authors declare no competing interests.
