## [Peer Review File · Communications Biology]

Reviewers' comments:

Reviewer #1 (Remarks to the Author):

In this manuscript, small-angle X-ray scattering (SAXS) has been used to determine the structure of the ROCK2 kinase on its own or in association with active RhoA. Consistent with previous data but in contrast with a recent structural study, the authors have determined that the coiled-coil region is flexible and allows the N-terminal kinase domain to associate with, and be inhibited by, the C-terminal Ph and Cysteine-rich domains. The binding of active RhoA helps to shift the structure away from the closed and inhibited conformation towards an extended and active conformation. In addition, large substrate proteins are presumed to also shift the conformation away from the closed inactive form towards the active state. The study is convincing and provides considerable substantiation to the autoinhibition mechanism of regulation, upon which some doubt had been cast. There are only a few minor points that should be considered.

1. As indicated on MS page 1, the ROCK proteins have been shown to be mainly located in the cytoplasm, although the C-terminal domains act to concentrate a proportion at the plasma membrane. An important role of active RhoA is likely to help increase the concentration of protein at the plasma membrane, as well as to promote the conformational shift towards the open conformation. Since this is the case, it seems likely that "free" ROCK dimers in the cytoplasm are in the closed conformation until such time that they encounter substrates that promote the adoption of the extended conformation. This would help to explain how they phosphorylate substrates that aren't necessarily associated with the cortical cytoskeleton, such as the LIM kinases, which are also cytoplasmic. This should be considered in the discussion of the manuscript.
2. On MS page 1, it would be helpful to indicate that MYPT1 is the subunit of the MLCP complex that is phosphorylated by ROCK. In addition, it should also be indicated that MLC II is also a substrate.
3. On MS page 2, the sizes of the ROCK kinases are presumably for the human forms?
4. On MS page 2, the discussion of the non-redundancy of the two ROCK proteins should also consider the paper Rho-associated Kinase (ROCK) Function Is Essential for Cell Cycle Progression, Senescence and Tumorigenesis by Kümper et al. (Elife 2016 Jan 14;5:e12994. doi: 10.7554/eLife.12203.), which concluded that the two proteins were redundant in fact.
5. On MS page 2, activation of ROCK1 by caspase-mediated removal of the inhibitory C-terminus was also reported by Coleman et al. (Membrane blebbing during apoptosis results from caspase-mediated activation of ROCK I. Nat Cell Biol. 2001 Apr;3(4):339-45. doi: 10.1038/35070009.)
6. On MS page 3, the first sentence would be clearer as "Although ROCK2 was initially identified as a Rho-associated protein..."
7. Figure 5 would be better if the individual points of the replicates were shown, and statistical significance analysed and indicated on the graphs.

Reviewer #2 (Remarks to the Author):

Review of COMMSBIO-20-1262-T

Summary

This manuscript addresses the question of how ROCK2 kinase, which plays an important role in cytoskeletal remodeling, can access membrane proximal and membrane distal substrates. The authors use a combination of biophysical and biochemical methods to make the following claims:

- ROCK2 forms a ternary complex with RhoA but
- RhoA does not activate ROCK2
- RhoA biases the conformation of ROCK2 to a more extended conformation that can sample membrane-distal locations and substrates
- ROCK2 is autoinhibited by its C-terminal regulatory domains

However, the study suffers from a number of fundamental flaws, notwithstanding the fact that the data presented by the authors do not robustly support any of their claims.

Major comments

Figure 1. The authors present size exclusion chromatography data to support a ternary complex of ROCK2 with RhoA. The problem is that the data unambiguously refute that conclusion. The authors observe no complex between BODIPY-GTP-RhoA and ROCK2, but somehow manage to conclude that a ternary complex is present because BODIPY-GTP binds more tightly to RhoA in the presence of ROCK2 (though no data is shown to support this conclusion and the fact remains that no stable ternary complex is observed). Furthermore, the use of a Superose 12 column is inappropriate given the size of dimeric ROCK2 (320 kDa), so it is questionable whether the ROCK2 protein is well behaved in this experiment (or just in the void where aggregated proteins would elute). Since this piece of data is critical for the next two major parts of the manuscript, the study is already irretrievably compromised. It should also be noted that the authors actually replicate the findings of Truebestein et al (Nat. Comms. 2016) regarding the lack of a ternary complex between ROCK2 and RhoA. This is still controversial, but the data are compelling.

Figure 2. The authors use small-angle X-ray scattering (SAXS) to investigate the solution structure of the complex between ROCK2 and RhoA. Since the data in Fig. 1 don't support the existence of a ternary complex, any conclusions from these experiments are flawed. The authors add a 1.5 molar excess of RhoA to their MBP-ROCK2 for SAXS measurements, which are not done using in-line SEC (the gold standard for SAXS measurements). This means that the SAXS measurements are a convolution of ROCK2 and RhoA, in a 1:1.5 ratio, tumbling independently (see Fig. 1) in solution. No useful information can be derived about the 'ternary complex' under these conditions. It should also be noted that full-length ROCK2 is actually a well-behaved protein and suitable for solution studies without any solubility tags such as MBP. The authors' difficulties in handling ROCK2 suggest problems with their purification conditions.

Figure 3-4. The authors perform computational modeling of their SAXS data. Modeling doesn't make sense unless the differences are real and can be demonstrated to be real robustly. Technical point: the range used for Guinier analysis shown in Fig S1 clearly shows a non-linear decay of the data points, which might indicate some aggregation in the sample (the authors already noted concentration-dependent behavioural problems with their ROCK2 protein). The authors should always show the Guinier fit, range of fit, and accompanying residuals to allow for proper interpretation of their data.

Figure 5. Kinase activity assays. The authors perform a series of activity assays using different ROCK2 substrates. The findings contradict much more sophisticated analyses of enzymatic activity (Truebestein et al., 2016). For example, the authors claim that ROCK2 is autoinhibited by its C-terminal regulatory domains on the basis that its activity is 56% compared to the kinase domain alone against LIMK1, supported by other published findings (which are also highly dubious). However, this is not in agreement with more careful biochemistry done on full-length, truncated, and kinase domain-only ROCK2, in which no differences were observed at all against recombinant MLC2 substrate (Truebestein et al., 2016). Furthermore, the authors then contradict their conclusion that the C-terminus exerts an autoinhibitory effect on the N-terminal kinase domains. In the case of the first experiment, they use LIMK1 as a substrate, in the second LIMK2, which are very similar (therefore the opposite finding doesn't make sense). The discrepancies in their findings probably reflect errors in their kinase assays, which are not very sophisticated single point assays. A further note on this point: the authors use fairly crude purification strategies for both their ROCK proteins and their substrates, as well as using an inappropriate SEC column (Superdex 75) for ROCK purification. It is therefore unclear just how pure their proteins are and, consequently, how accurately they can be quantitated. Finally, the authors conclude that RhoA exerts no effect on the activity of ROCK2, which is consistent with the findings of Truebestein et al (2016), but also entirely expected.

General

Figures are poorly annotated and difficult to interpret.

Reviewer #3 (Remarks to the Author):

The rho-associated protein kinases are proteins with many diverse and critical functions within the cell, and are also targets of pharmaceutical research, so understanding the gross conformational changes is important and of broad interest. The authors present protein models of full-length protein based on small-angle X-ray scattering data. The structures of sub-domains of ROCK1/2 are known, however the overall structure of the full-length protein is not known.

The key conclusions of the SAXS analysis are that the rho-binding and kinase domains are likely to be proximal in the absence of rho, and separated in the presence of rho. This has been long hypothesized, and the SAXS data supports this model.

The authors point out that the extended conformation observed by EM is likely entropically unfavorable. They favor ensembles of states with either greater or lesser radii of gyration (dependent). The difference is important as full length ROCK protein are often depicted in an extended, rigid conformation.

I would recommend the paper for publication (with changes indicated below).

Comments/edits:

(1) Page 2: "The enzymatic activity of ROCK kinases requires homodimerization(ref 16)..." is mis-stated. The authors of ref 16 paper actually conclude the opposite: "The fact that the monomers have similar turnover numbers as the dimers shows that dimerization is not essential for ROCK activity". I have no doubt that the dimerization is critical for cellular function, but it is not required for enzymatic activity. The authors should change this sentence to indicate that the cellular or biological activity requires dimerization and provide an additional reference.

(2) Page 3: clarify that the Rg measured is for MBP fusions: "The Rg values were 14.5 nm for the MBP-ROCK2 and 15.1 nm for the MBP-ROCK2/RhoA complex, indicating..." Is this correct (that MBP-fusions are the source of the Rg measurements)?

(3) Page 3, in ROCK2 structural models from SAXS: Again, this section should clarify that the protein in the model is an MBP-fusion. Are the Rg values of 15.8 and 11.8 from the model (with/without RhoA) consistent with the measured Rg values of 15.1 and 14.5 (with and without RhoA)?

-Marc Jacobs

Point-by-point answers:

Reviewer #1:

In this manuscript, small-angle X-ray scattering (SAXS) has been used to determine the structure of the ROCK2 kinase on its own or in association with active RhoA. Consistent with previous data but in contrast with a recent structural study, the authors have determined that the coiled-coil region is flexible and allows the N-terminal kinase domain to associate with, and be inhibited by, the C-terminal Ph and Cysteine-rich domains. The binding of active RhoA helps to shift the structure away from the closed and inhibited conformation towards an extended and active conformation. In addition, large substrate proteins are presumed to also shift the conformation away from the closed inactive form towards the active state. The study is convincing and provides considerable substantiation to the autoinhibition mechanism of regulation, upon which some doubt had been cast. There are only a few minor points that should be considered.

1. As indicated on MS page 1, the ROCK proteins have been shown to be mainly located in the cytoplasm, although the C-terminal domains act to concentrate a proportion at the plasma membrane. An important role of active RhoA is likely to help increase the concentration of protein at the plasma membrane, as well as to promote the conformational shift towards the open conformation. Since this is the case, it seems likely that “free” ROCK dimers in the cytoplasm are in the closed conformation until such time that they encounter substrates that promote the adoption of the extended conformation. This would help to explain how they phosphorylate substrates that aren’t necessarily associated with the cortical cytoskeleton, such as the LIM kinases, which are also cytoplasmic. This should be considered in the discussion of the manuscript.

We thank the reviewer for the comments and suggestions.

We have included a new sentence based on the suggestion of the reviewer in the discussion: *“Not only RhoA, but also the substrates of ROCK2 are probably capable of promoting the adoption of the extended conformation by forming protein-protein interactions with the loose closed-conformation forms of ROCK2.”* (page 5 lines 32-34)

2. On MS page 1, it would be helpful to indicate that MYPT1 is the subunit of the MLCP complex that is phosphorylated by ROCK. In addition, it should also be indicated that MLC II is also a substrate.

We have changed the sentence, and inserted a new reference regarding MLC II phosphorylation: *“Their most prominent targets are membrane-distal, localized at the actomyosin filaments, and include LIM kinases², the myosin phosphatase target subunit (MYPT) of myosin light chain phosphatase (MLCP)³, myosin light chain II (MLC-2)⁴, collapsin response mediator protein 2 (CRMP2)⁵, and ERM proteins⁶ (ezrin, radixin, and moesin)”.* (page 1 lines 27-30)

3. On MS page 2, the sizes of the ROCK kinases are presumably for the human forms?

We have added „the human” in the new sentence: *“The human ROCK1 and ROCK2 are highly homologous kinases consisting of 1354 and 1388 amino acids, respectively.”* (page 2 lines 3-4)

4. On MS page 2, the discussion of the non-redundancy of the two ROCK proteins should also consider the paper Rho-associated Kinase (ROCK) Function Is Essential for Cell Cycle Progression, Senescence and

Tumorigenesis by Kümpfer et al. (Elife 2016 Jan 14;5:e12994. doi: 10.7554/eLife.12203.), which concluded that the two proteins were redundant in fact.

We have modified the sentence, and added this reference (ref 16) in the manuscript: *"The functions of these kinases were initially considered highly overlapping, but knockout studies showed that disruption of either gene is sufficient to generate stillborn embryos^{14,15}, indicating that at least in early development, the two enzymes cannot substitute for each other, although it was shown that they can compensate for each other in adult organisms.¹⁶"* (page 2 lines 6-9)

5. On MS page 2, activation of ROCK1 by caspase-mediated removal of the inhibitory C-terminus was also reported by Coleman et al. (Membrane blebbing during apoptosis results from caspase-mediated activation of ROCK I. Nat Cell Biol. 2001 Apr;3(4):339-45. doi: 10.1038/35070009.)

We have added this reference (ref 21) to the manuscript.

6. On MS page 3, the first sentence would be clearer as "Although ROCK2 was initially identified as a Rho-associated protein..."

We have changed the sentence to *"Although ROCK2 kinase was initially identified as a Rho-associated protein, the direct binding between recombinant RhoA and ROCK2 was recently questioned²⁹, despite the fact that even a crystal structure of a complex between RhoA and a ROCK1 fragment had been published²⁵."* (page 3 lines 3-5)

7. Figure 5 would be better if the individual points of the replicates were shown, and statistical significance analysed and indicated on the graphs.

We have added the individual points of the replicates to the graphs, and calculated the statistical significance using a t-test.

Reviewer #2

This manuscript addresses the question of how ROCK2 kinase, which plays an important role in cytoskeletal remodeling, can access membrane proximal and membrane distal substrates. The authors use a combination of biophysical and biochemical methods to make the following claims:

- ROCK2 forms a ternary complex with RhoA but
- RhoA does not activate ROCK2
- RhoA biases the conformation of ROCK2 to a more extended conformation that can sample membrane-distal locations and substrates
- ROCK2 is autoinhibited by its C-terminal regulatory domains

We thank the reviewer for scrutinizing our manuscript and for the critical remarks. We have revised our manuscript to address the points of criticism as described below. First of all, we summarize our claims and their experimental support:

1. Rho kinase 2 (ROCK2) is an ensemble of partially folded conformations

This statement is robustly supported by SAXS experimental data.

2. ROCK2 forms a weak ternary complex with RhoA

The gel filtration and the SAXS experiments on ROCK2 – RhoA mixture support this.

3. RhoA does not activate ROCK2

This issue is controversial (according to the literature). Our kinase activity experiments unequivocally support the claimed view.

4. RhoA shifts the distribution of ROCK2 conformations towards more extended ones, enabling both membrane distal and proximal functions

This conclusion was drawn from SAXS experiments. We did consider the reviewer's argumentation. The detected weak interaction between ROCK2 and RhoA does not result in quantitative formation of a stable ROCK2/RhoA complex, therefore the ROCK2/RhoA "mixture" does not represent the pure complex. In the mixture of molecules with substantial size difference (408 kD vs. 20 kD) the scattering is dominated by the large molecules while the smaller makes a negligible impact on the scattering pattern (the calculated ratio of contribution is 38000:1 if the shape of the molecules is identical, and 400:1 even in the case of maximal deviation in the shapes). The observed increase in the gyration radius does indicate the appearance of more elongated structures in the presence of RhoA, i.e. a shift towards more extended conformations.

5. Autoinhibition of ROCK2 by its C-terminal domains is observed only in the case of small synthetic substrates

This conclusion is strongly supported by our kinase activity data.

Major comments

Figure 1. The authors present size exclusion chromatography data to support a ternary complex of ROCK2 with RhoA. The problem is that the data unambiguously refute that conclusion. The authors observe no complex between BODIPY-GTP-RhoA and ROCK2, but somehow manage to conclude that a ternary complex is present because BODIPY-GTP binds more tightly to RhoA in the presence of ROCK2 (though no data is shown to support this conclusion and the fact remains that no stable ternary complex is observed). Furthermore, the use of a Superose 12 column is inappropriate given the size of dimeric ROCK2 (320 kDa), so it is questionable whether the ROCK2 protein is well behaved in this experiment (or just in the void where aggregated proteins would elute). Since this piece of data is critical for the next two major parts of the manuscript, the study is already irretrievably compromised. It should also be noted that the authors actually replicate the findings of Truebestein et al (Nat. Comms. 2016) regarding the lack of a ternary complex between ROCK2 and RhoA. This is still controversial, but the data are compelling.

The reviewer suggests that our ROCK2 samples may be aggregated. To demonstrate that they are not, we have added the results of a gel filtration experiment obtained on Superose 6 (Figure S2).

The sample peak appears at approx. 1.6 MDa size, but considering the elongated shape of the ROCK2 molecule, this corresponds to the expected size. The shape of the peak is approximately symmetric, and the retention volume (10.5 ml) is significantly higher than the exclusion volume (7.6ml), showing that ROCK2 is not forming aggregates. Our SAXS data support the Mw value of 408 000 with an accuracy of 10%. Anti-ROCK2 and anti-MBP antibodies recognize the protein. Therefore, we believe that the doubt regarding the quality and identity of our ROCK2 samples is not justified. Reviewer #2 correctly points out that the Superose 12 column is inappropriate to identify the ROCK2 dimer, but in fact, we used this column for another purpose: to study the ratio of free BODIPY-GTP and BODIPY-GTP-RhoA complex (with sizes 950 D and 21kD).

We performed our gel filtration experiments with Superose 12 following the method of Blumenstein and Ahmadian (reference 30) to show the consequences of the complex formation indirectly. In the case of RhoA- BODIPY-GTP mixture, 20% of the fluorescence appears at the RhoA peak, while upon addition of ROCK2, this increases to 50 %. We can conclude that there is a ROCK2 and RhoA-(GTP) interaction i.e. ROCK2 and RhoA form a weak, transient complex. To facilitate interpretation of the binding experiments, we have designed a new Figure 1 based on the integration of the areas under the curves of the chromatograms. The original Figure 1 is attached as a Supplementary Figure.

Figure 2. The authors use small-angle X-ray scattering (SAXS) to investigate the solution structure of the complex between ROCK2 and RhoA. Since the data in Fig. 1 don't support the existence of a ternary complex, any conclusions from these experiments are flawed. The authors add a 1.5 molar excess of RhoA to their MBP-ROCK2 for SAXS measurements, which are not done using in-line SEC (the gold standard for SAXS measurements). This means that the SAXS measurements are a convolution of ROCK2 and RhoA, in a 1:1.5 ratio, tumbling independently (see Fig. 1) in solution. No useful information can be derived about the 'ternary complex' under these conditions. It should also be noted that full-length ROCK2 is actually a well-behaved protein and suitable for solution studies without any solubility tags such as MBP. The authors' difficulties in handling ROCK2 suggest problems with their purification conditions.

The referee is correct that the SAXS experiments do not demonstrate the existence of a stable ROCK2/RhoA complex. However, we do not make this claim. On the contrary, we interpret the SAXS and gel filtration results assuming a weak dynamic interaction between ROCK2 and RhoA shifted towards the dissociated form. We have refined the text to avoid this misunderstanding: *“Although we could not detect the ternary complex between ROCK2/RhoA/BODIPY-GTP directly, indirect evidence was obtained. BODIPY-GTP bound much more strongly to RhoA in the presence of ROCK2 than in its absence, suggesting the presence of a ternary complex similar to that measured for ROCK1³⁰, although this ternary complex is weak, and the dynamic equilibrium between ROCK2 and RhoA/BODIPY-GTP is probably shifted towards the dissociated form”*. (page 3 lines 8-13)

SAXS experiments show that the addition of RhoA to the solution of ROCK2 results in an increase ($14.63 \pm 0.18 \rightarrow 15.09 \pm 0.16$ nm) in the calculated R_g value, reflecting a loosening of the folded, flexible coiled-coil chain. The reviewer's argument about the individually tumbling ROCK2 and RhoA in the solution without interaction is not in accord with the observed increase in R_g . The addition of RhoA (Mw: 20.4 kDa) to the solution of partially folded ROCK2 (Mw: 408 kDa), would only result in a small R_g increase well within the limit of error if ROCK2 does not change its conformation. But the measured increase in R_g exceeds the experimental error, and supports, in accordance with the gel filtration results (Figure 1), the existence of an interaction between ROCK2 and RhoA, resulting in a population shift towards more relaxed conformations and extended distances between the termini.

The reviewer suggests that ROCK2 is a well-behaved protein in itself, thus the MBP tag would not have been necessary. While we agree with this statement, we found that the presence of the MBP tag increases the solubility of ROCK2 allowing us to use a higher protein concentration (5.4 μ M with MBP vs 3.8 μ M without MBP) in the SAXS experiments. As the precision of SAXS depends on protein concentration, we decided to use the MBP-linked forms in our experiments. The MBP-tag also helps to prevent radiation damage during the SAXS experiments.

Figure 3-4. The authors perform computational modeling of their SAXS data. Modeling doesn't make sense unless the differences are real and can be demonstrated to be real robustly. Technical point: the range used for Guinier analysis shown in Fig S1 clearly shows a non-linear decay of the data points, which might indicate some aggregation in the sample (the authors already noted concentration-dependent behavioural problems with their ROCK2 protein). The authors should always show the Guinier fit, range of fit, and accompanying residuals to allow for proper interpretation of their data.

In the revised version of the manuscript, we show the Guinier fit, range of fit, and accompanying residuals as requested (Supplementary figure 3). The slight upturn at the initial range of the

scattering curves was ascertained to be the result of parasitic scattering from the capillary, not the consequence of protein aggregation.

Figure 5. Kinase activity assays. The authors perform a series of activity assays using different ROCK2 substrates. The findings contradict much more sophisticated analyses of enzymatic activity (Truebestein et al., 2016). For example, the authors claim that ROCK2 is autoinhibited by its C-terminal regulatory domains on the basis that its activity is 56% compared to the kinase domain alone against LIMK1, supported by other published findings (which are also highly dubious). However, this is not in agreement with more careful biochemistry done on full-length, truncated, and kinase domain-only ROCK2, in which no differences were observed at all against recombinant MLC2 substrate (Truebestein et al., 2016). Furthermore, the authors then contradict their conclusion that the C-terminus exerts an autoinhibitory effect on the N-terminal kinase domains. In the case of the first experiment, they use LIMK1 as a substrate, in the second LIMK2, which are very similar (therefore the opposite finding doesn't make sense). The discrepancies in their findings probably reflect errors in their kinase assays, which are not very sophisticated single point assays. A further note on this point: the authors use fairly crude purification strategies for both their ROCK proteins and their substrates, as well as using an

inappropriate SEC column (Superdex 75) for ROCK purification. It is therefore unclear just how pure their proteins are and, consequently, how accurately they can be quantitated. Finally, the authors conclude that RhoA exerts no effect on the activity of ROCK2, which is consistent with the findings of Truebestein et al (2016), but also entirely expected.

Based on our results with the three natural substrates, we concluded that the activities of the full-length ROCK2 and its isolated kinase domain do not differ significantly; thus, we agree with the reviewer that no auto-inhibition occurs. To clarify our stance on this point, in the results section, we have reformulated the text as follows: *“On LIMK1, the activity of the full-length ROCK2 is 56% of that of the isolated kinase domain, while on the myosin binding subunit (MYPT1) of myosin phosphatase and LIMK2, the measured activity of the full length kinase was higher (by 19% and 11% respectively) than that of the kinase domain. This difference is close to the standard deviation of the experiment, so we can make the qualitative statement that no apparent inhibition occurs in the presence of the C-terminal domains.”* (page 4 lines 18-22)

The proteins were purified to >90% purity based on SDS-PAGE analysis. The Superdex 75 column was used for gel filtration of the ROCK2 kinase domain, while the full-length ROCK2 was gel filtrated on Superose 6. We corrected this in the “Methods” section of the manuscript: *“The eluted protein was applied to a Superose 6 (ROCK2-full) or Superdex 75 (ROCK2-KD) gel filtration column equilibrated in 25mM HEPES pH 7.4, 150mM NaCl, 1mM DTT.”* (page 6 lines 31-32)

General

Figures are poorly annotated and difficult to interpret.

We have made changes to the figures for better annotation and interpretation. We have converted Fig. 2 into colour, the annotation of Fig. 4 has been improved, and Fig. 1 has been completely redesigned.

Reviewer #3:

The rho-associated protein kinases are proteins with many diverse and critical functions within the cell, and are also targets of pharmaceutical research, so understanding the gross conformational changes is important and of broad interest. The authors present protein models of full-length protein based on small-angle X-ray scattering data. The structures of sub-domains of ROCK1/2 are known, however the overall structure of the full-length protein is not known.

The key conclusions of the SAXS analysis are that the rho-binding and kinase domains are likely to be proximal in the absence of rho, and separated in the presence of rho. This has been long hypothesized, and the SAXS data supports this model.

The authors point out that the extended conformation observed by EM is likely entropically unfavorable. They favor ensembles of states with either greater or lesser radii of gyration (dependent). The difference is important as full length ROCK protein are often depicted in an extended, rigid conformation.

I would recommend the paper for publication (with changes indicated below).

We thank the reviewer for the review and the suggestions.

(1) Page 2: "The enzymatic activity of ROCK kinases requires homodimerization (ref 16)..." is mis-stated. The authors of ref 16 paper actually conclude the opposite: "The fact that the monomers have similar turnover numbers as the dimers shows that dimerization is not essential for ROCK activity". I have no doubt that the dimerization is critical for cellular function, but it is not required for enzymatic activity. The authors should change this sentence to indicate that the cellular or biological activity requires dimerization and provide an additional reference.

We have made several changes in this paragraph, and added a new reference (ref 13): *"The enzymatic activity of ROCK kinases is strongly enhanced by homodimerization¹⁷ where the N- and C-terminal extensions of the kinase domain function as a dimerization interface in addition to the coiled-coil helical regions. The dimerization of the protein is suggested to play a significant role in its biological function towards dimeric substrates¹³."* (page 2 lines 13-17)

(2) Page 3: clarify that the R_g measured is for MBP fusions: "The R_g values were 14.5 nm for the MBP-ROCK2 and 15.1 nm for the MBP-ROCK2/RhoA complex, indicating..." Is this correct (that MBP-fusions are the source of the R_g measurements)?

We have clarified in the revised manuscript that these values were determined for the MBP fusions: *"The R_g values were 14.5 nm for the MBP-ROCK2 alone and 15.1 nm for MBP-ROCK2 in the presence of RhoA, indicating some loosening of the structure. The experimental molecular masses (458 and 490 kDa for MBP-ROCK2 and MBP-ROCK2/RhoA, respectively) agreed with the calculated masses of the dimeric MBP-ROCK2 (408 and 450 kDa) within the margin of experimental error."* (page 3 lines 25-29)

(3) Page 3, in ROCK2 structural models from SAXS: Again, this section should clarify that the protein in the model is an MBP-fusion. Are the R_g values of 15.8 and 11.8 from the model (with/without RhoA) consistent with the measured R_g values of 15.1 and 14.5 (with and without RhoA)?

Here, we have also clarified that the model is for an MBP-fusion: *“We have obtained a good fit between the experimental scattering curves and those calculated from the best models obtained by geometric simulations (Fig. 2). In the best model, we have obtained for the MBP-ROCK2 fusion protein without bound RhoA, the chain is folded in half and one N-terminal domain is in contact with a C-terminal domain (Fig. 3a). In the best model obtained for the RhoA-bound MBP-ROCK2, the molecule is less compact (R_g : 15.8/11.8 nm with/without RhoA), in agreement with the experimentally detected increase of R_g complexed with RhoA, and the termini are not in proximity (Fig. 3b).”* (page 3 lines 35-37; continued on page 4 lines 1-4)

We have also added an explanation for the differences between the experimental R_g values and those calculated from the models: *“The discrepancies between the R_g values obtained from the scattering data and those calculated from the structural models are due to the fact that the experimental R_g is calculated from a small stretch of the Guinier plot while the structural models are scored based on the whole scattering curve; also, the experimental value is for a solution ensemble of structures while the value obtained from the structural model is for a single structure.”* (page 4 lines 4-8)

--

REVIEWERS' COMMENTS:

Reviewer #3 (Remarks to the Author):

This review is for the revised manuscript.

I reviewed my own prior comments as well as the other two reviewers. The comments from Reviewer #2 were the most focused on the technical details of the protein preparation, interpretation of SEC data, SAXS data, and the kinase assays.

The revised manuscript addresses the reviewer comments through the addition of clarified text, additional text, additional references, clarified figures, and some supplemental data. I believe that the revised manuscript addresses the concern of Reviewer #2 from my point of view, as well as my own proposed changes.